# Antibody-Mediated In Vitro Activation and Expansion of Blood Donor-Derived Natural Killer Cells with Transient Anti-Tumor Efficacy

**DOI:** 10.3390/biomedicines13122934

**Published:** 2025-11-29

**Authors:** Shengxue Luo, Feifeng Zeng, Qitao Deng, Yalin Luo, Dawei Chen, Hui Ren, Wenjie Xia, Xin Ye, Shuxin Huang, Tingting Li, Yongshui Fu, Xia Rong, Huaqin Liang

**Affiliations:** 1Institute of Blood Transfusion and Hematology, Guangzhou Blood Center, Guangzhou Medical University, Guangzhou 510095, China; shengxueluo@163.com (S.L.); 18701265960@163.com (F.Z.); luoyalin922@163.com (Y.L.); 428dawei@163.com (D.C.); renhui1995@126.com (H.R.); wendy060428leon@126.com (W.X.); 779684055@qq.com (X.Y.); hsxshu@163.com (S.H.); fuyongshui@sina.com (Y.F.); 2The Key Medical Laboratory of Guangzhou, Guangzhou 510091, China; 3Department of Transfusion Medicine, School of Laboratory Medicine and Biotechnology, Southern Medical University, Guangzhou 510515, China; dengqitao@i.smu.edu.cn (Q.D.); apple-ting-007@163.com (T.L.); 4Guangzhou First People’s Hospital, South China University of Technology, Guangzhou 510180, China

**Keywords:** peripheral blood mononuclear cells, super NK cells, anti-CD16, anti-CD137, anti-tumor efficacy

## Abstract

**Background:** Natural killer (NK) cells are key effectors of innate immunity with broad-spectrum anti-tumor activity. However, peripheral blood-derived NK (PBNK) cells are typically quiescent, which limits their therapeutic utility. This study aimed to develop an efficient strategy for the in vitro activation and expansion of PBNK cells and then evaluate their potential anti-tumor efficacy in vitro and vivo. **Methods:** NK cells were isolated from healthy blood donors’ peripheral blood and stimulated with anti-CD16 and anti-CD137 antibodies in the presence of interleukin-2 (IL-2) and interleukin-15 (IL-15) under serum-free conditions, generating super NK (SNK) cells. The expression levels of activating and inhibitory receptors on the expanded SNK cells were assessed by flow cytometry. Cytotoxicity against tumor cells was assessed at various effector-to-target (E:T) ratios in vitro. In vivo, anti-tumor efficacy was evaluated in K562-engrafted NSG mice. RNA sequencing was performed to identify differentially expressed genes (DEGs) between SNK and PBNK cells. **Results:** Stimulation with anti-CD16 and anti-CD137 antibodies resulted in significant expansion of donor-derived NK cells, with over 861.9 ± 48.84-fold expansion (n = 5) within 15 days of culture. SNK cells exhibited significantly elevated expression of activating receptors, including NKG2D. Functionally, SNK cells demonstrated superior cytotoxicity compared with PBNK cells across all tested E:T ratios in vitro and higher expressions of the effector molecules interferon-gamma (IFN-γ) and granzyme B (Gzm B). In vivo, adoptive SNK cell transfer resulted in significant tumor suppression and prolonged survival in a dose-dependent manner. Transcriptomic analysis revealed significant enrichment of DEGs associated with cytokine and chemokine signaling, immune activation, and cytotoxic effector function compared with the PBNK cells. **Conclusions:** Anti-CD16/CD137 antibody stimulation, in combination with IL-2 and IL-15, facilitates robust activation and rapid expansion of functionally enhanced NK cells from peripheral blood. The resulting SNK cells demonstrated enhanced anti-tumor efficacy both in vitro and in vivo and may be used as allogeneic NK cell-based immunotherapy in future cancer treatment strategies.

## 1. Introduction

Natural killer (NK) cells are a vital component of peripheral blood mononuclear cells (PBMCs) [1]. Functionally mature NK cells from healthy blood donors express a diverse repertoire of activating and inhibitory receptors, enabling their broad-spectrum anti-tumor and anti-infection activity [2,3,4,5]. Unlike T cells, NK cells can recognize and eliminate virus-infected or tumor cells without prior antigen sensitization and are not restricted by the major histocompatibility complex, allowing for rapid immune responses [6,7]. Furthermore, their inability to induce graft versus host disease makes them a safe and universal “off-the-shelf” allogeneic immunotherapy suitable for diverse patient populations [8,9,10,11]. Despite these advantages, peripheral blood-derived NK cells (PBNK) are predominantly quiescent and exhibit limited anti-tumor activity. In contrast, in vitro-activated super NK (SNK) cells express high levels of activating receptors and exhibit significantly enhanced cytotoxicity against tumor cells [12,13].

The cytotoxic effect of NK cells is mediated primarily through receptor–ligand interaction involving natural cytotoxicity receptors (such as NKp30, NKp44, and NKp46) [14,15], activating receptors (such as NKG2D) [16], and activating killer cell immunoglobulin-like receptors that recognize specific ligands overexpressed on tumor cells [14,15,16]. This interaction triggers NK cell activation, leading to the release of cytotoxic granules containing perforin, granzymes, and granulysin, which induce apoptosis or lysis of target cells [17,18,19,20,21,22]. Furthermore, activated NK cells secrete pro-inflammatory cytokines, such as interferon-gamma (IFN-γ), and chemokines that help modulate both innate and adaptive immune responses [23].

PBMCs are often discarded during routine blood processing, leading to the loss of a potentially valuable immunotherapeutic resource [24,25]. Harnessing these cells offers a promising strategy for developing clinically relevant NK cell-based therapies. Current protocols for in vitro NK cell activation and expansion generally fall into two categories: feeder-free and feeder-dependent systems. Feeder-free methods, which depend on cytokine stimulation (e.g., IL-2, IL-15, or IL-21), typically result in limited expansion folds. Nevertheless, the resulting NK cells often exhibit heightened cytotoxic function [26,27,28,29]. In contrast, the use of irradiated feeder cells (e.g., K562-based artificial antigen-presenting cells) can achieve substantially higher expansion efficiencies, with reports of over 40,000-fold increases. However, this approach carries potential risks, including the unintended proliferation of feeder cells and contamination of the final NK cell product. Current in vitro NK cell activation and expansion methods carry a risk of feeder cell proliferation and contamination [29,30,31,32,33]. Therefore, it is necessary to develop safe and efficient methods for the activation and expansion of NK cells.

To address these limitations, this study employed an antibody of a CD16- and CD137-coated culture system to activate and expand NK cells from blood donor PBMCs under serum-free conditions. NK cell differentiation gives rise to major subsets, including CD56^bright^CD16^dim^ and CD56^dim^CD16^bright^ cells [34]. The CD16 receptor (FcγRIIIa) is central to the anti-tumor function of NK cells, as it triggers antibody-dependent cellular cytotoxicity (ADCC) upon interacting with IgG. This activation underpins a broader cytotoxic response, which involves pathways like Fas/FasL and cytokine release [35]. CD137, as a member of the tumor necrosis factor receptor superfamily, transmits co-stimulatory signals upon antibody binding, which promotes NK cell proliferation, survival, and effector function [36]. In combination with IL-2 and IL-15 [26,27,28,29], stimulation with CD16 and CD137 antibodies synergistically induces NK cell activation and proliferation, enabling their large-scale expansion under serum-free conditions to clinically relevant cell loads for adoptive transfer. The expansion dynamics, phenotypic characteristics, and anti-tumor efficacy of the expanded NK cells were comprehensively evaluated in both in vitro and in vivo models.

## 2. Materials and Methods

### 2.1. Cells and Animals

Human nasopharyngeal carcinoma cell line HK-1 (YC-C095, Ubigene, Cambridge, MA, USA), human prostate cancer cell line DU145 (HTB-81, ATCC, Manassas, VA, USA), and chronic myelogenous leukemia cell line K562 (CCL-243, ATCC, Manassas, VA, USA) were cultured in complete RPMI-1640 medium supplemented with 10% fetal bovine serum (FBS; Gibco, Grand Island, NY, USA) at 37 °C in a humidified atmosphere containing 5% CO_2_. All cell lines were lentivirally transduced to express firefly luciferase (Luc).

Female NOD-SCID-gamma (NSG) mice (6–8 weeks) were obtained from GemPharmatech (Nanjing, Jiangsu, China). All procedures involving animals were approved by the Southern Medical University Animal Care and Use Committee (Approval No. SMUL2023080 and 10 May 2023) and conducted according to the institutional guidelines for the care and use of laboratory animals.

### 2.2. Isolation and Expansion of NK Cells

Five independent buffy coats, obtained as byproducts of routine blood component separation, were used for PBMC isolation. A 4 mL aliquot of buffy coat was diluted with an equal volume of phosphate-buffered saline (PBS) and layered onto 10 mL of lymphocyte separation medium. Following centrifugation at 400× *g* for 25 min at room temperature, PBMCs were obtained, and PBNK cells were isolated using an NK cell isolation kit (130-092-657, Miltenyi Biotec, Bergisch Gladbach, Germany). The purified PBNK cells (2 × 10^5^ cells/well) were seeded into 24-well plates pre-coated with anti-CD16 (bsm-30017M, Bioss, Beijing, China), anti-CD137 (bs-23572R, Bioss, Beijing, China), or a combination of both antibodies (concentrations of 5 ug/mL per antibody). Cells were cultured in 2 mL of serum-free NK cell medium (130-114-429, Miltenyi Biotec, Bergisch Gladbach, Germany) supplemented with IL-2 (400 IU/mL; HZ-1015, Proteintech, Rosemont, IL, USA) and IL-15 (200 IU/mL; HZ-1337, Proteintech, Rosemont, IL, USA). Cell counts were recorded on days 5, 7, 9, 11, 13, and 15 to generate a proliferation curve.

### 2.3. Flow Cytometry

The phenotypic profiles of activated SNK cells were assessed using flow cytometry. PBNK and SNK cells (n = 3 donors/group) were collected and stained with fluorescently labeled monoclonal antibodies, including FITC-conjugated anti-human CD3 (555332, BD, Franklin Lake, NJ, USA) and APC-conjugated anti-human CD56 (555518, BD, Franklin Lake, NJ, USA). Activating and inhibitory markers were assessed using PE-conjugated antibodies against CD69, CD94, NKG2C, NKG2D, NKp46, NKp44, Tim-3, and CD96 (Appendix A). Stained cells were acquired on a BD FACSCanto flow cytometer (BD, Franklin Lake, NJ, USA), and data were analyzed using FlowJo software (version 10, Ashland, OH, USA) to determine the frequencies of activating and inhibitory marker expression.

### 2.4. Luc-Based Cytotoxicity Assays

The cytotoxic activity of PBNK and SNK cells was evaluated using a Luc-based cytotoxicity assay. PBNK or SNK cells were co-cultured with Luc-expressing target tumor cell lines (HK-1, DU145, and K562) at effector-to-target (E:T) ratios of 5:1, 2:1, 1:1, and 0.5:1 for 6 h (n = 3 per group). Luminescence was measured and expressed as relative light units (RLU). The percentage of tumor cell death was calculated using the following formula: Killing effect (%) = 100 × (RLU of untreated target cells − RLU of target cells co-cultured with NK cells)/RLU of untreated target cells.

### 2.5. Enzyme-Linked Immunosorbent Assay (ELISA)

Culture supernatants from the cytotoxicity assays were collected after 6 h of co-culture with target cells (n = 3 per group). The concentrations of interferon-γ (IFN-γ) and granzyme B (Gzm B) were determined using human IFN-γ (EK180HS-96, MultiSciences, Hangzhou, China) and Gzm B ELISA kits (EK158-96, MultiSciences, Hangzhou, China), according to the manufacturer’s instructions.

### 2.6. Animal Grouping and Intervention

To establish an in vivo leukemia model, NSG mice were intravenously injected with 1 × 10^6^ K562-Luc cells through the tail vein. On day 3 post-tumor cell inoculation, mice were randomly assigned into four groups (n = 5 per group): three treatment groups, administered intravenous injections of SNK cells at doses of 2 × 10^7^, 4 × 10^7^, or 8 × 10^7^ cells in 200 µL of PBS, and a control group, administered 200 µL of PBS alone. Tumor burden was assessed weekly using a bioluminescence imaging system (IVIS Spectrum, Revvity, Waltham, MA, USA) following SNK administration. The body weight and general health status of mice were monitored daily to assess treatment-related effects.

### 2.7. RNA Sequencing

Total RNA was isolated from PBNK and SNK cells with TRIzol reagent (T9424, Sigma, St. Louis, MO, USA). Sequencing libraries were constructed using the Optimal Dual-mode mRNA Library Prep Kit (Beijing Genomics Institute, Beijing, China). Following cDNA synthesis and end repair, the libraries were amplified by PCR, quality-controlled, and denatured to yield single-stranded DNA. The single-stranded DNA was then circularized, and any remaining linear molecules were digested. The resulting DNA nanoballs were sequenced on a BGI T7 platform (Beijing Genomics Institute, Beijing, China) to generate 100/150 bp paired-end reads. Data analysis was conducted on the Dr. Tom multi-omics data mining system (Beijing Genomics Institute, Beijing, China). Gene expression levels were quantified with RSEM (v1.3.1), and a heatmap was generated using pheatmap (v1.0.12) to visualize expression patterns. Differential expression analysis was performed with DESeq2 (v1.34.0), applying a significance threshold of Q value ≤ 0.05.

### 2.8. Statistical Analyses

All statistical analyses and graphical representations were performed using GraphPad Prism software version 9 (San Diego, CA, USA). Data are presented as the mean ± standard error of the mean (SEM) from at least three independent biological replicates (unless otherwise stated), where a biological replicate is defined as cells derived from a unique healthy donor. Technical replicates were averaged to provide a single value for each biological replicate prior to statistical analysis. Intergroup comparisons were performed using either an unpaired Student’s *t* test or a one-way analysis of variance (ANOVA) followed by Bonferroni’s post hoc test, as appropriate. Mice survival curves were assessed using the log-rank Mantel–Cox test, and the survival benefit in the SNK-treated group was highly significant (*p* < 0.05, log-rank test), with a large effect size (Hazard Ratio = 2.59). Statistical significance was defined as follows: *****
*p* < 0.05, ******
*p* < 0.01, and *******
*p* < 0.001; ns = not significant.

## 3. Results

### 3.1. Anti-CD16 and Anti-CD137 Antibodies Induce Robust Activation and Expansion of Donor-Derived Human PBMCs

PBMCs were isolated from buffy coats, a byproduct of blood component preparation (Figure 1A). Subsequently, PBNK cells were enriched from the PBMCs through cell sorting, achieving a purity of over 98%, with the majority of cells being CD56^+^ (Figure 1B). To induce large-scale expansion, PBNK cells were stimulated with anti-CD16, anti-CD137, or both. Stimulation with anti-CD16 or anti-CD137 induced robust activation of PBNK cells compared with the unstimulated controls (Figure 1C). Notably, PBNK cells from five independent buffy coats were co-stimulated with anti-CD16 and anti-CD137 antibodies and resulted in the most pronounced proliferative response (Figure 1C), with over 861.9 ± 48.84-fold expansion (n = 5) by day 15 of culture, whereas stimulation with anti-CD16 or anti-CD137 alone resulted in approximately 214.7 ± 14.94-fold or 393.4 ± 23.83-fold expansion, respectively (Figure 1D). On day 15, the purity of the expanded NK population (SNK cells) in the co-stimulated group remained >98%, with predominant expression of CD56 (Figure 1E). These results demonstrate that combined stimulation of PBNK cells with anti-CD16 and anti-CD137 antibodies induces robust activation and large-scale expansion, yielding a high-purity population of SNK cells.

### 3.2. Elevated Expression of Activating and Inhibitory Receptors in SNK Cells

Flow cytometric analysis was performed to compare the expression of surface receptors associated with NK cell activation and inhibition between PBNK and SNK cells. The results revealed significant upregulation of both activating and inhibitory receptors in SNK cells compared with PBNK cells (Figure 2). CD69, an early activation marker, was expressed in 91.40 ± 4.78% of SNK cells, a substantial increase compared with 15.17 ± 5.04% observed in PBNK cells (Figure 2A), which suggests robust activation post-antibody stimulation. The inhibitory receptor CD94 was also significantly upregulated in SNK cells compared with PBNK cells (Figure 2B). Furthermore, the activating receptors NKG2C and NKG2D were significantly upregulated in SNK cells (Figure 2C,D). Other activating and inhibitory receptors, including NKp44, NKp46, Tim-3, and CD96, were also upregulated (Appendix A). These findings indicate that the activation and expansion of PBNK cells generate SNK cells with a mature, functionally active phenotype, characterized by elevated expression of both activating and inhibitory receptors, which may enhance their immunoregulatory and anti-tumor functions.

### 3.3. SNK Cells Exhibited Enhanced Cytotoxicity and Effector Molecule Secretion In Vitro

The cytotoxic potential of SNK cells was assessed using a Luc-based cytotoxicity assay against the HK-1, DU145, and K562 tumor cell lines (Figure 3A). Compared with PBNK cells, SNK cells exhibited significantly increased cytotoxicity against all three tumor cell lines after 6 h of co-culture at various E:T ratios (Figure 3B). At an E:T ratio of 1:1, SNK cells induced approximately 36.61 ± 5.92%, 37.27 ± 6.93%, and 48.41 ± 3.51% tumor cell (HK-1, DU145, and K562) lysis, compared with only 16.35 ± 3.78% induced by PBNK cells (Figure 3B). At an E:T ratio of 2:1, SNK cells elicited obviously cytotoxicity, inducing tumor cell (HK-1, DU145, and K562) lysis of 65.33 ± 1.45%, 73.21 ± 0.88%, and 99.14 ± 0.57%, respectively. Similar results were obtained with SNK cells from two other healthy blood donors (Appendix A). Notably, SNK cells eradicated approximately 99.14% of K562 cells, a significantly higher killing effect compared with that of PBNK cells (*p* < 0.001; Figure 3B). The cytotoxic activity of SNK cells also increased in a dose-dependent manner with higher E:T ratios (Figure 3C).

To further investigate the mechanism of this enhanced cytotoxicity, the levels of key effector molecules (IFN-γ and Gzm B) were measured by ELISA. The concentrations of IFN-γ (more than 573.72 ± 25.77 pg/mL; Figure 3D) and Gzm B (more than 1708.09 ± 48.47 pg/mL; Figure 3E) were significantly higher in the culture supernatant of SNK cells than in PBNK cells at an E:T ratio of 5:1 and the control group (tumor cells: 22.64 ± 5.95 pg/mL and SNK cells: 44.60 ± 4.53 pg/mL) (*p* < 0.05; Figure 3D,E). These findings indicate that SNK cells possess superior cytotoxic capacity and enhanced secretion of effector molecules, underscoring their anti-tumor potential in vitro.

### 3.4. SNK Cells Demonstrated Enhanced Anti-Tumor Efficacy In Vivo

To further evaluate the therapeutic potential of SNK cells in vivo, NSG mice were inoculated intravenously with 1 × 10^6^ K562-Luc tumor cells. Three days post-inoculation, mice were administered a single intravenous dose of SNK cells (2 × 10^7^, 4 × 10^7^, or 8 × 10^7^ cells) through tail vein injection (Figure 4A). Tumor progression was monitored weekly by bioluminescence imaging, while body weight and survival were recorded daily. The results revealed that SNK cell treatment significantly inhibited tumor growth in a dose-dependent manner compared with PBS controls, with the most pronounced suppression observed on day 7 post-treatment (day 10 post-K562-Luc inoculation) (Figure 4B). Survival analysis revealed that higher doses of SNK cells significantly prolonged survival compared with PBS controls (median survival: 19 days vs. 17 days, *p* = 0.032; Figure 4C). Despite initial tumor suppression, rapid tumor proliferation was observed after day 13, accompanied by significant weight loss in mice (Figure 4D). In addition, the survival time of SNK cell treatment mice was detected by flow cytometry in the peripheral blood (PB) and spleen (Figure 4E–G). In PB, the percentage of SNK cells reached the highest ratio (14.65 ± 0.84%, n = 5 mice) on day 7 post-SNK infusion (day 10 post-K562 infusion), then decreased to a low level of 2.32±0.17% (n = 4 mice) by day 13 post-SNK infusion (day 16 post-K562 infusion) and 1.06±0.19% (n = 4 mice) by day 15 post-SNK infusion (day 18 post-K562 infusion) (Figure 4E,F). In the spleen, the percentage of SNK cells reached the highest level of 16.83±0.78% (n = 3 mice) on day 7 post-SNK infusion (day 10 post-K562 infusion), then decreased a low level of 1.97 ± 0.16% (n = 4 mice) by day 15 post-SNK infusion (day 18 post-K562 infusion) (Figure 4G). This late-phase tumor regrowth is likely attributed to the reduced persistence or viability of the transferred SNK cells after days 7-10 (Figure 4E–G), leading to a reduction in their inhibitory effect on the tumor. Collectively, these findings suggest that SNK cells suppress tumor progression and prolong survival in vivo in a dose-dependent manner. A single infusion of SNK cells led to significant suppression of tumor growth until approximately day 10, after which tumor relapse was observed in the model. However, sustained tumor control may require additional strategies to enhance the persistence of SNK cells.

### 3.5. Enhanced Cytokine Signaling and Immune Activation Signatures in SNK Cells

To investigate the molecular mechanisms underlying SNK cell activation, RNA sequencing was performed to compare the gene expression profiles of SNK cells stimulated with anti-CD16 and anti-CD137 antibodies with those of unstimulated PBNK cells.

A total of 5350 differentially expressed genes (DEGs) were identified, with 2708 transcripts upregulated and 2642 downregulated in SNK cells (Figure 5A). Kyoto Encyclopedia of Genes and Genomes (KEGG), Reactome, and Gene Ontology (GO) enrichment analyses revealed that the upregulated genes in SNK cells were primarily associated with cell cycle regulation, DNA replication, and DNA repair (Figure 5B–D). Additionally, a heatmap of DEGs showed that SNK cells exhibited significantly increased expression of genes (Figure 5E–H). Violin plots of the KEGG analysis showed upregulation of genes related to cytokine and chemokine signaling (including CCR2, CCR4, CCR5, CCL23, CXCR6, and CXCL13; Figure 5I), immune activation (including CD48, CD70, and GNB4; Figure 5J), cytotoxic effector function (including GZMB, NCR2, NCR3, TNFrsf10, TNFrsf12, and PRF1; Figure 5K), and signaling by interleukin (including IL2RA, IL2RG, IL12RB2, and IL32; Figure 5L) in the SNK cell group. These transcriptomic profiles suggest that CD16 and CD137 activation induces broad transcriptional reprogramming in SNK cells, which promotes the expression of genes associated with proliferation, immune signaling, and cytotoxicity and potentially contributes to the enhanced anti-tumor effects of SNK cells.

## 4. Discussion

NK cells are key effectors of the innate immune system, capable of recognizing and eliminating tumor- or virus-infected cells through direct cytolysis and secretion of immunomodulatory cytokines. Their crucial role in anti-tumor immunity has been widely documented [7,11]. However, the clinical translation of NK cell-based immunotherapies is limited by the low abundance of functional PBNK cells and challenges associated with their large-scale expansion [12,13]. In this study, an optimized approach for the activation and expansion of NK cells from healthy donor peripheral blood was developed using antibodies targeting CD16 and CD137 in a serum-free medium. This method generated a high-purity population of SNK cells, which demonstrated significant anti-tumor activity in both in vitro and in vivo models.

Stimulation with anti-CD16 and anti-CD137 antibodies resulted in the robust activation and expansion of PBNK cells under serum-free conditions, achieving over 861.9 ± 48.84-fold expansion (n = 5) by day 15 post-stimulation (Figure 1D). The purity of the resulting SNK cells was >98%, with >94% expressing a CD56^bright^ phenotype (Figure 1E). Flow cytometric analysis further revealed significant upregulation of activating receptors (Figure 2), including CD69, NKG2D, NKG2C, NKp44, and NKp46, as well as inhibitory receptors, such as CD94-NKG2A, Tim-3, and CD96, in SNK cells compared with unstimulated PBNK cells, indicating the acquisition of a mature and functionally active phenotype.

Mechanistically, current protocols for NK cell activation and expansion frequently rely on cytokines, particularly IL-2 and IL-15 [28]. While IL-2 is critical for promoting NK cell proliferation and boosting cytotoxicity and IL-15 supports differentiation and counters apoptosis [26,27], a primary drawback of this cytokine-based approach is its insufficient potency in activating resting NK cells. Consequently, cultures dependent solely on IL-2 and IL-15 frequently exhibit suboptimal proliferative capacity and slow expansion, a challenge accentuated in serum-free systems [28,29]. CD16, an Fcγ receptor (FcγRIIIa), mediates NK cell activation by binding to the Fc portion of IgG, triggering immune receptor tyrosine activation motif-dependent signaling that promotes cytotoxicity and cytokine release [35]. CD137 (also known as 4-1BB/TNFrsf9), a member of the tumor necrosis factor receptor superfamily, transmits co-stimulatory signals upon antibody binding, which promotes NK cell proliferation, survival, and effector function [36]. In combination with IL-2 and IL-15 [27,28,29,30,31,32,33,34,35,36], stimulation with CD16 and CD137 antibodies synergistically induces NK cell activation and proliferation, enabling their large-scale expansion under serum-free conditions to clinically relevant cell loads for adoptive transfer.

Functionally, SNK cells demonstrated superior cytotoxicity against all three tumor cell lines in vitro, with particularly pronounced effects against the K562 cell line (Figure 3). At an E:T ratio of 2:1, SNK cells achieved complete lysis of K562 cells within 6 h post-culture (Figure 3B,C). This enhanced cytotoxicity was accompanied by significantly increased secretion of effector molecules, including IFN-γ and Gzm B. In vivo, SNK cells significantly inhibited K562 tumor growth and prolonged survival in NSG mice in a dose-dependent manner, compared with PBS controls (Figure 4). However, tumor regrowth was observed after day 13, likely reflecting limited SNK cell persistence (Figure 4E–G). Therefore, achieving a more profound survival benefit will likely require a multi-dosing strategy to maintain effective SNK cell levels in vivo.

Transcriptomic profiling further revealed that SNK cells exhibited DEGs involved in immune activation, cytokine and chemokine signaling, cytotoxicity, and cell cycle regulation, identified candidate genes, and suggested potential mechanisms of PBNK cells through the antibodies CD16 and CD137 (Figure 5). These transcriptional signatures may suggest that CD16 and CD137 antibody activation induces broad reprogramming of NK cells toward a proliferative, highly cytotoxic, and immunoregulatory phenotype.

Limitations: the anti-tumor efficacy of SNK cells in vivo was only evident in the K562 leukemia model for immunodeficient NSG mice, and the lack of data on solid tumors precludes an assessment of the ability of SNK cells to overcome tumor microenvironment barriers. The artificial nature of the immunodeficiency model precludes the possibility of studying the interaction of allogeneic NK cells with the recipient’s immune system, including potential rejection or alloreactivity. Using only the K562 cell line, which is highly sensitive to NK cells, may exaggerate the effectiveness of the SNK cells against tumors in vivo. Without data on solid tumor models and immune-competent systems, claims of broad anti-tumor potential for SNK cells appear premature. Therefore, further investigating the activation mechanisms of PBNK cells by antibodies CD16 and CD137 and evaluating the efficacy and safety profiles of SNK cells in humanized mice models are warranted.

## 5. Conclusions

In summary, this study has established a robust, serum-free method for the activation and large-scale production of highly pure, functionally potent NK cells using anti-CD16 and anti-CD137 antibody stimulation. The resulting SNK cells demonstrated an activated immunophenotype and potent cytotoxicity against tumor cells both in vitro and in vivo. Furthermore, this study has highlighted the need for additional strategies to improve the in vivo persistence of SNK cells to achieve long-term tumor control.

## Figures and Tables

**Figure 1 biomedicines-13-02934-f001:**
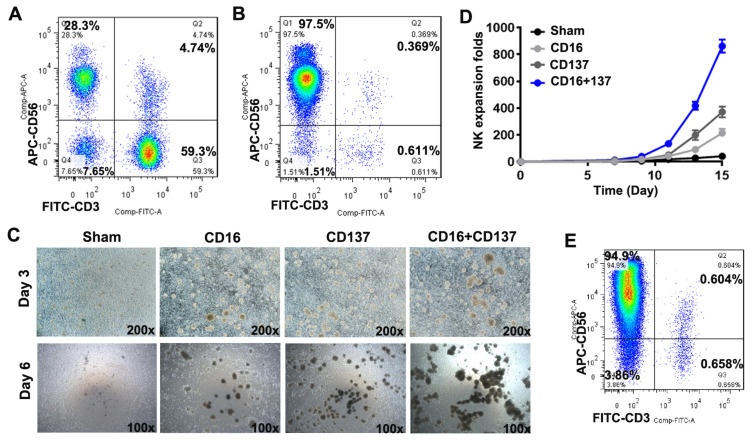
Isolation, activation, and expansion of peripheral blood-derived NK cells (CD56^+^ CD3^−^). (**A**) PBMCs were isolated from blood donor buffy coats and analyzed by flow cytometry using anti-CD3 and anti-CD56 antibodies. (**B**) The purity of the isolated PBNK cells was determined by flow cytometry. (**C**) Activation of PBNK cells with anti-CD16 and/or anti-CD137 antibodies. (**D**) Expansion folds of PBNK cells after stimulation (n = 5 donors). (**E**) The purity of the expanded SNK cells was analyzed by flow cytometry on day 15 post-stimulation.

**Figure 2 biomedicines-13-02934-f002:**
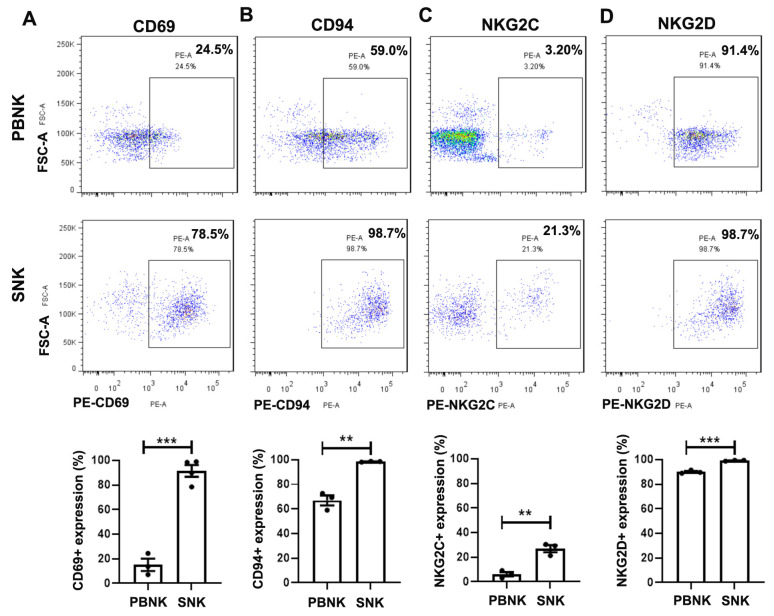
Elevated expression of activating and inhibitory receptors in SNK cells compared with PBNK cells (n = 3 donors). PBNK and SNK cells were stained with anti-CD3 and anti-CD56 antibodies, followed by receptor-specific antibodies (PE-positive, black box). The frequencies of CD69 (**A**), CD94 (**B**), NKG2C (**C**), and NKG2D (**D**) were determined by flow cytometry. Data are presented as mean ± SEM. Statistical analysis was performed using an unpaired Student’s *t* test. Significant differences are indicated as ** *p* < 0.01 and *** *p* < 0.001.

**Figure 3 biomedicines-13-02934-f003:**
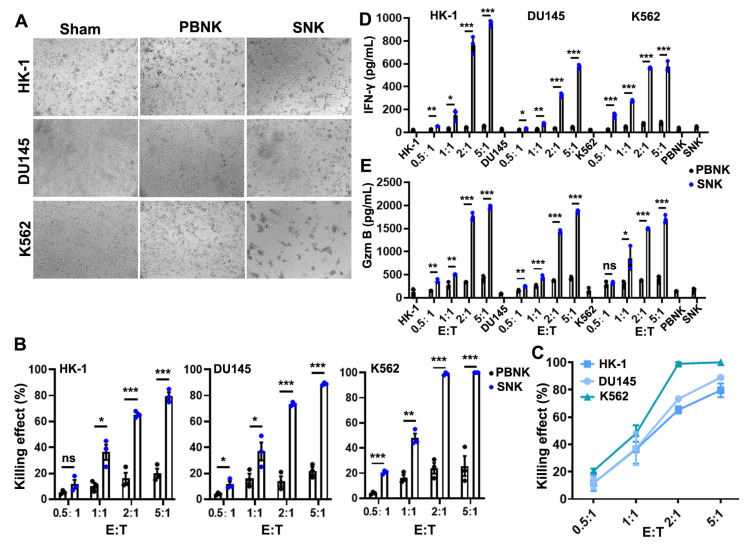
SNK cells exhibited enhanced cytotoxicity and effector molecule secretion in vitro (n = 3 donors). (**A**) Representative images depicting the cytotoxic effects of SNK and PBNK cells against HK-1, DU145, and K562 tumor cells after 6 h of co-culture. (**B**) Cytotoxicity was quantified at various E:T ratios using luciferase-based assays (for the other two donors of SNK against tumors, see also Appendix A). (**C**) Mean cytotoxicity against all three tumor cells, determined by luminescence. (**D**,**E**) Concentrations of IFN-γ (**D**) and Gzm B (**E**) in the co-culture supernatants were quantified by ELISA. Data are presented as mean ± SEM. Statistical analysis was performed using Student’s *t* test. Significant differences are indicated as * *p* < 0.05, ** *p* < 0.01, and *** *p* < 0.001; ns = not significant.

**Figure 4 biomedicines-13-02934-f004:**
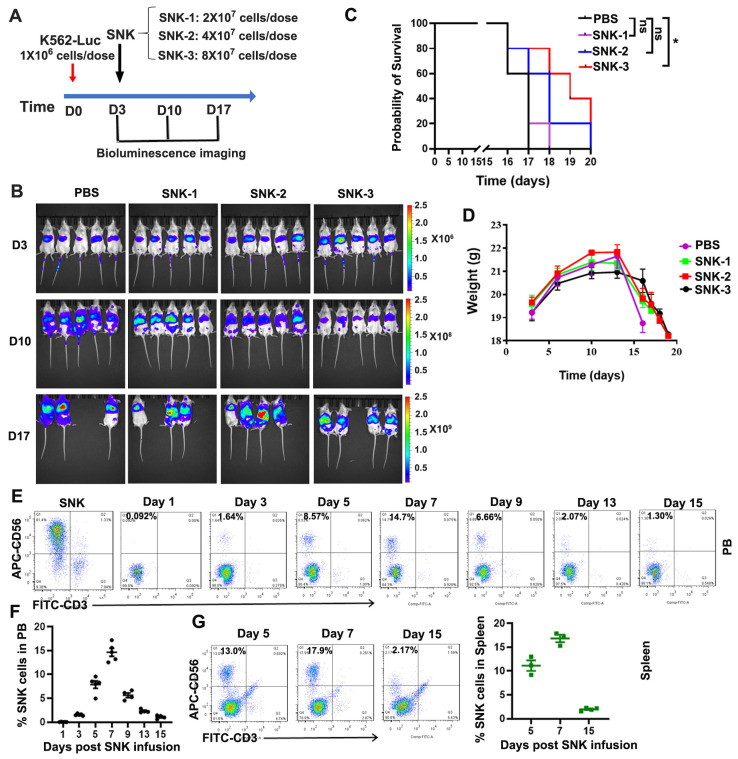
SNK cells exhibited enhanced cytotoxicity against tumor cells in vivo. (**A**) NSG mice were intravenously injected with 1 × 10^6^ K562-Luc cells through the tail vein. On day 3, mice were intravenously administered SNK cells at doses of 2 × 10^7^, 4 × 10^7^, or 8 × 10^7^ cells. Tumor burden was monitored on days 3, 10, and 17 using bioluminescence imaging. (**B**) Representative bioluminescence images showing tumor progression over time. (**C**) Kaplan–Meier survival curves were generated, and statistical significance was determined using the log-rank test (* *p* < 0.05; ns = not significant), with a large effect size (Hazard Ratio = 2.59). (**D**) The body weights of mice were measured post-K562 cell inoculation at the indicated time points. The percentage of SNK cells (CD56^+^ CD3^−^) was determined by flow cytometry in the peripheral blood (PB, (**E**,**F**)) and spleen (**G**) of mice post-SNK cell infusion.

**Figure 5 biomedicines-13-02934-f005:**
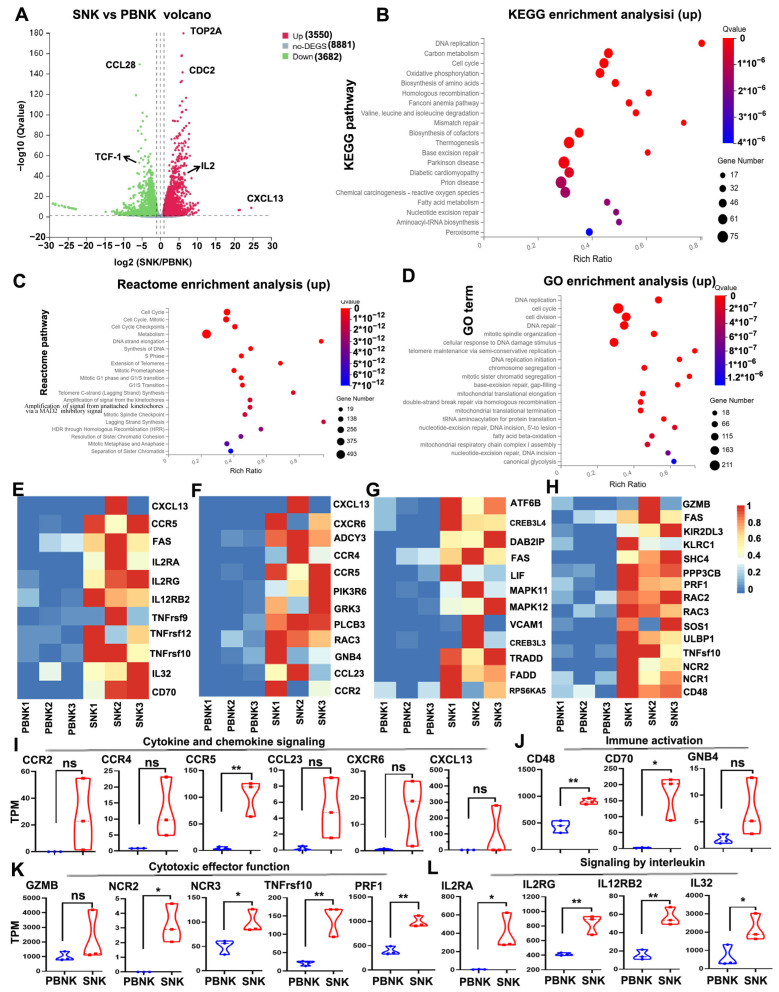
Transcriptomic profiling showing DEGs between SNK and PBNK cells. (**A**) Volcano plot showing DEGs between SNK and PBNK cells. (**B**) KEGG, (**C**) Reactome, and (**D**) GO enrichment analyses showing upregulated genes in SNK cells. (**E**–**H**) Heatmaps showing DEGs associated with cytokine and chemokine signaling (**E**), immune activation (**F**), and cytotoxicity (**G**,**H**). (**I**–**L**) Violin plots showing the transcripts per million (TPM) of cytokine and chemokine signaling, immune activation, cytotoxic effector function, and signaling by interleukin-related genes in PBNK and SNK cell groups (n = 3 donors). Statistical analysis was performed using an paired Student’s *t* test. Significant differences are indicated as * *p* < 0.05 and ** *p* < 0.01; ns = not significant.

## Data Availability

All data generated or analyzed during this study are included in this published article and its Appendix A.

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
