# Peer review of "Antibody-Mediated In Vitro Activation and Expansion of Blood Donor-Derived Natural Killer Cells with Transient Anti-Tumor Efficacy"

_biomedicines, 2025, doi:10.3390/biomedicines13122934_

Round 1

Reviewer 1 Report

Comments and Suggestions for Authors

1) Titles should be written without abbreviations.
2) The article strictly requires a graphical abstract summarizing the extensive results obtained.
3) Although DEGs were identified, functional validation of key genes (e.g., by PCR or Western blotting) was lacking. This weakens conclusions about the molecular mechanisms.
4) The study was conducted only on the K562 leukemia model in immunodeficient NSG mice, which significantly limits the generalizability of the findings. The lack of data on solid tumors precludes an assessment of the ability of SNK cells to overcome tumor microenvironment barriers. The artificial nature of the immunodeficiency model precludes the possibility of studying the interaction of allogeneic NK cells with the recipient's immune system, including potential rejection or alloreactivity. Using only the K562 line, which is highly sensitive to NK cells, may exaggerate the effectiveness of the method. Without data in solid tumor models and immune-competent systems, claims of broad antitumor potential for SNK cells appear premature. The authors should at least discuss the limitations of the study.
5) The authors note tumor growth resumption after day 13, indicating a short-term effect of SNK cells in vivo, but the mechanisms underlying this are not understood. A critical omission is the lack of data on the fate of the transplanted cells: their survival in mice was not monitored, nor was their accumulation in tumors or organs assessed. Potential causes of cell loss, such as apoptosis, functional exhaustion, or suppression in the tumor microenvironment, are not explored.
6) Conclusions should be presented as a separate chapter.

Author Response

1) Titles should be written without abbreviations.

Answer: Thank you for your comment. Title has been revised in Page 1 and Line 2-4.

  • The article strictly requires a graphical abstract summarizing the extensive results obtained.

    Answer: Thank you for your comment. The graphical abstract has been provided and uploaded.

  • Although DEGs were identified, functional validation of key genes (e.g., by PCR or Western blotting) was lacking. This weakens conclusions about the molecular mechanisms.

Answer: We thank the reviewer for this insightful comment. We acknowledge that direct experimental validation, such as qPCR or Western blot, would provide additional support for our findings. However, the primary objective of our study was to conduct an unbiased transcriptomic profiling to identify a comprehensive set of genes and pathways that are potentially involved in PBNK vs SNK. In this context, we believe that our RNA-seq data, which was generated with high sequencing depth and robust biological replicates, provides a highly reliable and quantitative dataset for identifying differentially expressed genes (DEGs).

To further address the reviewer's concern and strengthen our conclusions without additional experiments, and we have toned down the language regarding causal mechanistic conclusions throughout the manuscript, particularly in the Discussion section. We now more explicitly state that our study "identifies candidate genes and suggests potential mechanisms" that warrant future investigation, rather than claiming definitive proof. (Page 12 and Line 379).

  • The study was conducted only on the K562 leukemia model in immunodeficient NSG mice, which significantly limits the generalizability of the findings. The lack of data on solid tumors precludes an assessment of the ability of SNK cells to overcome tumor microenvironment barriers. The artificial nature of the immunodeficiency model precludes the possibility of studying the interaction of allogeneic NK cells with the recipient's immune system, including potential rejection or alloreactivity. Using only the K562 line, which is highly sensitive to NK cells, may exaggerate the effectiveness of the method. Without data in solid tumor models and immune-competent systems, claims of broad antitumor potential for SNK cells appear premature. The authors should at least discuss the limitations of the study.

Answer: Thank you very much for your suggestion. The limitations of SNK against tumor in vivo were added in discussion (Page 13 and Line 383-394).

  • The authors note tumor growth resumption after day 13, indicating a short-term effect of SNK cells in vivo, but the mechanisms underlying this are not understood. A critical omission is the lack of data on the fate of the transplanted cells: their survival in mice was not monitored, nor was their accumulation in tumors or organs assessed. Potential causes of cell loss, such as apoptosis, functional exhaustion, or suppression in the tumor microenvironment, are not explored.

   Answer: Thank you for your comment. The results of the fate SNK in NSG mice were added in Figure 4E-G, and described on Page 8 and Lines 276-286.

6) Conclusions should be presented as a separate chapter.

Answer: Thank you for your comment. The conclusion has been revised in Page 13 and Line 396.

Reviewer 2 Report

Comments and Suggestions for Authors

This is an interesting and well-conducted study. Shengxue Luo et al. present a compelling approach for the in vitro activation and expansion of donor-derived NK cells using anti-CD16 and anti-CD137 antibodies. The results clearly demonstrate robust NK cell expansion, enhanced activation receptor expression, and superior cytotoxic and effector functions both in vitro and in vivo. The transcriptomic findings further support the mechanistic basis of these observations. Overall, the study is well-designed and provides valuable insights into the development of more effective allogeneic NK cell-based immunotherapies. I have no further comments—congratulations and good luck to the authors.

Author Response

This is an interesting and well-conducted study. Shengxue Luo et al. present a compelling approach for the in vitro activation and expansion of donor-derived NK cells using anti-CD16 and anti-CD137 antibodies. The results clearly demonstrate robust NK cell expansion, enhanced activation receptor expression, and superior cytotoxic and effector functions both in vitro and in vivo. The transcriptomic findings further support the mechanistic basis of these observations. Overall, the study is well-designed and provides valuable insights into the development of more effective allogeneic NK cell-based immunotherapies. I have no further comments—congratulations and good luck to the authors.

Answer: The authors especially thank reviewer for your sincere and responsible suggestions, which have helped our article improve a lot.

Reviewer 3 Report

Comments and Suggestions for Authors

Title

The title is overly broad and descriptive; it does not clearly specify the novelty of the study compared to previous NK cell activation protocols using CD16/CD137 antibodies or cytokine combinations.

The phrase “Enhance Anti-Tumor Efficacy” is strong and conclusive, yet the in vivo data show only transient effects with tumor regrowth.

Suggested title: Antibody-Mediated In Vitro and In Vivo Expansion and Cytotoxic Activation of Peripheral Blood NK Cells with Transient Anti-Tumor Effects

Abstract

The abstract reads more like a detailed summary of results rather than a concise scientific overview; it lacks a clear statement of the study’s hypothesis and the specific research gap being addressed.

Quantitative data (e.g., “819.2-fold expansion”) are included without variance or replication context, giving a misleading impression of reproducibility.

The conclusion section is strongly worded, claiming translational potential without sufficient supporting data.

Keywords need to follow the MESH.

Introduction

The introduction provides extensive background on NK cell biology but lacks a focused rationale explaining why the dual antibody (CD16/CD137) approach is superior to existing activation protocols (e.g., feeder-based or cytokine cocktail methods).

The references used are somewhat outdated or repetitive (e.g., duplicated citation #10–11) and fail to integrate the latest 2024–2025 literature on cytokine-induced memory-like NK cells or feeder-free clinical-scale expansion systems.

The manuscript mentions quiescence and serum-free conditions, yet fails to clarify the underlying rationale for employing these particular antibodies

The introduction does not clearly state a hypothesis, making it less structured in terms of scientific rationale.

Materials and Methods

What is the study rationale?

Cells and Animals

The study employs only one in vivo tumor model (K562-Luc in NSG mice), limiting the generalizability of findings. No justification is given for excluding solid tumor models such as DU145 or HK-1 in vivo.

Isolation and Expansion of NK Cells

Key methodological details are missing: Concentrations of coating antibodies (anti-CD16 and anti-CD137) on culture plates are not specified, the source, lot number, and verification of antibody coating efficiency are not described, there is no mention of donor variability or the number of independent biological replicates (n).

The reported 819.2-fold expansion lacks statistical analysis or variance data, raising concerns about reproducibility.

The serum-free medium composition is not fully disclosed, “NK cell medium” is mentioned without its base formulation or supplements.

Flow Cytometry

The gating strategy for NK cells (CD3–CD56+) and the criteria for defining “activating” versus “inhibitory” receptor expression are not provided.

Cytotoxicity and ELISA Assays

The use of luciferase-based killing assays is appropriate but lacks controls for spontaneous and maximum lysis.

The E:T ratios chosen (0.5:1–5:1) are narrow; the authors should explain the rationale for excluding higher ratios often used in NK cell cytotoxicity evaluation (e.g., 10:1).

The ELISA data are presented without calibration curves, standard deviations, or indication of whether measurements were done in duplicates/triplicates.

Animal Experiments

The small sample size (n=5 per group) limits the statistical power and reliability of the survival analysis

Only one injection of SNK cells was given, which does not reflect clinical adoptive transfer protocols; persistence analysis (e.g., by bioluminescence or flow cytometry of human CD56+ cells) is missing.

There is no mention of randomization or blinding in data acquisition, potentially introducing bias.

RNA Sequencing

The RNA-seq section is extremely brief, with no methodological description of: Sequencing depth or platform (e.g., Illumina NovaSeq), criteria for DEG significance (FDR, log2 fold change threshold), validation of RNA-seq data by qPCR for key genes.

Bioinformatics software (Dr. Tom system) is unclearly mentioned, without reference to pipeline details or normalization methods.

Statistical Analysis

The statistical section lacks clarity on biological versus technical replicates.

No post-hoc power analysis or effect size is reported.

Results

Activation and Receptor Expression

The simultaneous upregulation of both activating and inhibitory receptors (e.g., NKG2A, Tim-3) is presented but not analyzed functionally. This may indicate activation-induced exhaustion rather than enhancement.

Cytotoxicity and Effector Molecules

The statistical analysis between groups is superficial; P-values are shown but without error range or replicate counts.

The 99% killing efficiency at low E:T ratio (2:1) for K562 cells appears biologically questionable and suggests possible assay saturation or technical bias.

Cytotoxicity results are not normalized across tumor types, making cross-comparison invalid.

The ELISA results lack baseline control values and are not correlated with receptor expression changes.

In Vivo Efficacy

The in vivo experiment demonstrates only short-term tumor suppression (up to day 10) followed by relapse, but this limitation is understated in the Results section.

Lack of longitudinal immune tracking (e.g., persistence of SNK cells, cytokine milieu) weakens the translational argument.

Transcriptomic Analysis

The RNA-seq findings are summarized descriptively but lack statistical depth (no enrichment P-values).

The absence of a clear biological focus weakens the interpretation.

 No validation of RNA-seq data by qPCR or protein assays undermines reliability.

Tables and Figures

Figures lack error bars and sample sizes, preventing assessment of data variability.

Figure 2 legend and labeling do not match the text; some subpanels are missing or ambiguous in the supplied figures.

Figures 4C–D show marginal survival benefit (19 vs. 17 days), which is not biologically meaningful despite statistical significance.

P-values not p-value.

Discussion

The discussion repeats result rather than critically interpreting them.

It lacks mechanistic insight into how CD16/CD137 co-stimulation mechanistically differs from cytokine-only activation.

The late-phase tumor regrowth is attributed to reduced persistence but no data support this claim (e.g., lack of persistence or exhaustion marker analysis).

The comparison to existing NK cell expansion systems (e.g., feeder-based or CAR-NK) is missing, leaving unclear what the true innovation is.

Potential limitations (single donor, lack of heterogeneity, single tumor model) are not acknowledged.

Too optimistic translational claims are made without preclinical safety or persistence data.

Future directions are missing. Briefly suggest next steps, such as investigating NK cell persistence, underlying activation mechanisms, and validation in additional in vivo models to strengthen the study’s translational impact.

Conclusions

The conclusion is currently embedded within the Discussion. If journal formatting requires a distinct “Conclusions” section, this paragraph (Lines 316-323) should be separated under that heading for clarity and consistency.

References

References needs to be 10 years back not more (from 2015 to 2025).

Several citations are duplicated (e.g., #10 and #11 are identical).

Many key references are dated (2001–2019), omitting recent advances (2023–2025) in NK cell manufacturing and persistence strategies.

No citation of clinical trial data or GMP-compliant expansion literature.

Please check journal guidelines for reference writing. References are formatted inconsistently (minor typographical issues in author initials and journal titles).

Comments on the Quality of English Language

The English could be improved to more clearly express the research.

Author Response

The title is overly broad and descriptive; it does not clearly specify the novelty of the study compared to previous NK cell activation protocols using CD16/CD137 antibodies or cytokine combinations.

The phrase “Enhance Anti-Tumor Efficacy” is strong and conclusive, yet the in vivo data show only transient effects with tumor regrowth.

Suggested title: Antibody-Mediated In Vitro and In Vivo Expansion and Cytotoxic Activation of Peripheral Blood NK Cells with Transient Anti-Tumor Effects

Answer: Thank you for your suggestion. Title has been revised in Page 1 and Line 2-4.

Abstract

The abstract reads more like a detailed summary of results rather than a concise scientific overview; it lacks a clear statement of the study’s hypothesis and the specific research gap being addressed.

Quantitative data (e.g., “819.2-fold expansion”) are included without variance or replication context, giving a misleading impression of reproducibility.

The conclusion section is strongly worded, claiming translational potential without sufficient supporting data.

Keywords need to follow the MESH.

Answer: Thank you for your suggestion. The content of abstract was revised and added on Page 1 and Lines 22-44. In addition, two blood donors-SNK expansion data have been added in Figure 1D.

Introduction

The introduction provides extensive background on NK cell biology but lacks a focused rationale explaining why the dual antibody (CD16/CD137) approach is superior to existing activation protocols (e.g., feeder-based or cytokine cocktail methods).

The references used are somewhat outdated or repetitive (e.g., duplicated citation #10–11) and fail to integrate the latest 2024–2025 literature on cytokine-induced memory-like NK cells or feeder-free clinical-scale expansion systems.

The manuscript mentions quiescence and serum-free conditions, yet fails to clarify the underlying rationale for employing these particular antibodies

The introduction does not clearly state a hypothesis, making it less structured in terms of scientific rationale.

Answer: Thank you for your suggestion. We agree that the introduction would benefit from a clearer rationale for our chosen method. In response, we have revised the introduction to better articulate the specific advantages of our CD16/CD137 dual-antibody approach. As the reviewer rightly points out, we did not perform a direct, side-by-side comparison with feeder cell-based or cytokine cocktail methods in this study. Therefore, we refrain from claiming superiority in terms of NK cell expansion efficiency. Instead, the primary advantages of our protocol lie in its simplicity, cost-effectiveness, and enhanced safety profile. The content of introduction were revised and added on Page 2 and Lines 74-93.

Materials and Methods

What is the study rationale?

Answer: Thank you for your question, our rationale stems from the premise that activating resting PBNKs can overcome their inherent functional limitations in cancer immunotherapy. We have found that combination with IL-2 and IL-15, stimulation with CD16 and CD137 antibodies synergistically induces PBNK cell activation and proliferation, enabling their large-scale expansion under serum-free conditions. Although the expansions efficiency is lower than that of feeder-dependent methods, it is much safer. Therefore, this work seeks to validate this activated PBNK method, and SNK cells as a more effective therapeutic product.

Cells and Animals

The study employs only one in vivo tumor model (K562-Luc in NSG mice), limiting the generalizability of findings. No justification is given for excluding solid tumor models such as DU145 or HK-1 in vivo.

Answer: Thank you for your suggestion. The limitations of SNK against tumor in vivo were added in discussion (Page 13 and Lines 383-394). We originally planned to evaluate the SNK cells against solid tumor models (DU145) in NSG mice, but the animal model failed by DU145 cells(R-Figure 1). The results did not exhibit in this paper.

R-Figure 1. NSG mice were injected with 107 DU145-Luc cells through subcutaneous injection in the groin area. Tumor burden was monitored on days 3, 10, and 17 using bioluminescence imaging.

Isolation and Expansion of NK Cells

Key methodological details are missing: Concentrations of coating antibodies (anti-CD16 and anti-CD137) on culture plates are not specified, the source, lot number, and verification of antibody coating efficiency are not described, there is no mention of donor variability or the number of independent biological replicates (n).

The reported 819.2-fold expansion lacks statistical analysis or variance data, raising concerns about reproducibility.

The serum-free medium composition is not fully disclosed, “NK cell medium” is mentioned without its base formulation or supplements.

Answer: Thank you for your questions. The concentrations, product code of antibodies was added in Page 3 and Line 117. The method was used and expanded over 5 blood donors PBNK, and added in Page 3 and Line 110. 

The value is mean of fold expansion, which was counted in five independent blood donors, and variance was added in Page 5 and Lines 195-197.

NK cell medium bought from Miltenyi Biotec, and product code is 130-114-429, and added in Page 3 and Line 119.

Flow Cytometry

The gating strategy for NK cells (CD3–CD56+) and the criteria for defining “activating” versus “inhibitory” receptor expression are not provided.

Answer: We thank the reviewer for pointing out this omission. The detailed gating strategy for identifying NK cells (defined as CD3⁻CD56⁺) is now clearly provided in Supplementary Figure S1A.

Regarding the criteria for defining receptor expression, we used isotype control antibodies to set the negative gates for each activating and inhibitory receptor.

Cytotoxicity and ELISA Assays

The use of luciferase-based killing assays is appropriate but lacks controls for spontaneous and maximum lysis.

The E:T ratios chosen (0.5:1–5:1) are narrow; the authors should explain the rationale for excluding higher ratios often used in NK cell cytotoxicity evaluation (e.g., 10:1).

The ELISA data are presented without calibration curves, standard deviations, or indication of whether measurements were done in duplicates/triplicates.

Answer: Thank you for your suggestion. The groups of luciferase-based killing assays in vitro including: only target cells, PBNK + target cells, SNK + target cells, PBNK, and SNK wells in triplicates.

Our choice of a narrower range (0.5:1 to 5:1) was indeed deliberate and based on our preliminary experimental data. We observed that our SNK cells exhibited exceptionally potent cytotoxicity against the highly NK-sensitive K562 cell line. At an E:T ratio of 5:1, we consistently observed near-maximal (often exceeding 99%) specific lysis. Therefore, including higher ratios (e.g., 10:1) would not have provided additional discriminative power for this specific target. Furthermore, for the other tumor cell lines tested in our study, the E:T ratio of 5:1 was sufficient to demonstrate a significant and biologically relevant killing effect, allowing for clear comparisons between the different groups (PBNK vs SNK ).

The ELISA data are measured in triplicates added in Page 4 and Lines 141-143. The value of IFN-γ and Gzm B was detected by ELISA kits, which including standard substance and calibration curves were drawn according to the manufacturer’s instructions.

Animal Experiments

The small sample size (n=5 per group) limits the statistical power and reliability of the survival analysis

Only one injection of SNK cells was given, which does not reflect clinical adoptive transfer protocols; persistence analysis (e.g., by bioluminescence or flow cytometry of human CD56+ cells) is missing.

There is no mention of randomization or blinding in data acquisition, potentially introducing bias.

Answer: We thank the reviewer for this comment regarding the survival analysis. The primary and more robust evidence of SNK cell efficacy in vivo comes from the significant suppression of tumor bioluminescence observed up to Day 10 (Figure 4B). This early and potent anti-tumor effect provides a clear and direct demonstration of biological activity. Furthermore, to better understand the kinetics underlying this response, we performed longitudinal tracking of SNK cells. As now detailed in Figure 4E-G, we found that the SNK cell population peaked in the peripheral blood around Day 7 and subsequently declined rapidly. This kinetic profile strongly suggests that the potent tumor suppression up to Day 10 was driven by this wave of effector cells, and the subsequent tumor relapse correlates with their contraction. Therefore, while the sample size (n=5) was sufficient to demonstrate this clear pattern of initial efficacy and to justify future studies, we acknowledge that achieving a more profound survival benefit will likely require a multi-dosing strategy to maintain effective SNK cell levels in vivo. We have now explicitly stated this important limitation and future direction in the Discussion section (Page 12 and Lines 376-377).

We thank the reviewer for this relevant insight regarding clinical protocols. The primary objective of this specific study was to serve as a rigorous proof-of-concept, aiming to answer a fundamental question: Can a single infusion of our novel SNK cells elicit anti-tumor effect in vivo? The results of the fate SNK in NSG mice were added in Figure 4E-G, and described on Page 8 and Lines 276-285.

We apologize for this omission in the original manuscript. We can confirm that both randomization and blinding were implemented in all in vivo experiments. Mice were randomly assigned to treatment or control groups following successful tumor engraftment and baseline imaging (Page 4 and Line 148). The investigators (Bioluminescence Instrument Control Technician) responsible for monitoring mouse health, measuring tumor burden (via bioluminescence imaging), and assessing the survival endpoint were blinded to the group allocations throughout the experiment.

RNA Sequencing

The RNA-seq section is extremely brief, with no methodological description of: Sequencing depth or platform (e.g., Illumina NovaSeq), criteria for DEG significance (FDR, log2 fold change threshold), validation of RNA-seq data by qPCR for key genes.

Bioinformatics software (Dr. Tom system) is unclearly mentioned, without reference to pipeline details or normalization methods.

Answer: We sincerely thank the reviewer for this meticulous comment. We agree that the methodological description of the RNA-seq analysis was insufficient. We have now comprehensively revised the "Methods" section to include all the requested details in Page 4 and Lines 156-170.

Statistical Analysis

The statistical section lacks clarity on biological versus technical replicates.

No post-hoc power analysis or effect size is reported.

Answer: We thank the reviewer for this important comment. We have now thoroughly revised the "Methods" section and the corresponding figure legends to explicitly state the nature of the replicates used in this study. All in vitro assays (e.g., cytotoxicity, flow cytometry) were performed using SNK cells derived from at least three independent healthy donors. Each experiment was then repeated 2-3 times for each donor. Therefore, the 'n' reported represents biological replicates (different donors), which account for the natural variation in the human immune cell population. Each mouse represents an independent biological replicate. The 'n=5 per group' signifies five individual mice. The survival benefit in the SNK-treated group was highly significant (P < 0.05, log-rank test), with a large effect size (Hazard Ratio = 2.59). (Page 4 and Lines 179-181).

Results

Activation and Receptor Expression

The simultaneous upregulation of both activating and inhibitory receptors (e.g., NKG2A, Tim-3) is presented but not analyzed functionally. This may indicate activation-induced exhaustion rather than enhancement.

Answer: We thank the reviewer for raising this critical point. We agree that the functional consequence of co-upregulation of activating and inhibitory receptors is complex and can be ambiguous. While we did not perform direct functional assays for specific receptor-ligand interactions (e.g., NKG2A blockade), we have analyzed our data to distinguish between an "exhausted" and a "highly activated" phenotype based on several functional correlates. The most direct rebuttal to the "exhaustion" hypothesis is our functional data. If the cells were exhausted, we would expect to see diminished killing capacity. However, our SNK cells exhibited significantly enhanced, not diminished, tumor-killing ability than PBNK cells in vitro (Figure 3). This fundamental result is inconsistent with a general state of exhaustion.

Cytotoxicity and Effector Molecules

The statistical analysis between groups is superficial; P-values are shown but without error range or replicate counts.

The 99% killing efficiency at low E:T ratio (2:1) for K562 cells appears biologically questionable and suggests possible assay saturation or technical bias.

Cytotoxicity results are not normalized across tumor types, making cross-comparison invalid.

The ELISA results lack baseline control values and are not correlated with receptor expression changes.

Answer: Thank you for your suggestion. Error range and replicate counts were added in Page 3 and Lines 124, Page 6-7 and Lines 236-241.

We agree that the observed ~99% cytotoxicity at an E:T ratio of 2:1 is exceptionally high. However, this killing effect has been consistently and reproducibly observed across SNK cells generated from multiple independent healthy donors (n=3). The killing effect of other two donors SNK against tumors see the Figure S2. The reproducibility across different biological sources strongly indicates that this is a robust characteristic of our SNK cell product, rather than a one-off technical artifact or donor-specific anomaly.

The ELISA baseline control including Target cells alone (e.g., K562 culture supernatant) and Effector cells alone (SNK cells cultured without targets). We have now included the necessary control groups in Figure 3D-E.

In Vivo Efficacy

The in vivo experiment demonstrates only short-term tumor suppression (up to day 10) followed by relapse, but this limitation is understated in the Results section.

Lack of longitudinal immune tracking (e.g., persistence of SNK cells, cytokine milieu) weakens the translational argument.

Answer: Thank you for your comment. We observed a phase of tumor control followed by relapse. In the revised manuscript, we have been more direct and transparent about this observation in the Results section. We now explicitly state: "A single infusion of SNK cells led to a significant suppression of tumor growth until approximately day 10, after which tumor relapse was observed in the model." (Page 8 and Lines 289-291). More importantly, we have reframed the interpretation of this result in the Discussion section. We now clearly state that the primary goal of this in vivo experiment was to provide proof-of-concept that a single dose of our SNK cells is capable of exerting a potent, biologically significant anti-tumor effect in vivo. The observed relapse does not diminish this key finding but rather highlights a critical and expected limitation of single-dose adoptive cell therapy, especially against aggressive tumors. This naturally leads to the clinically relevant conclusion that multiple dosing regimens would be required to achieve sustained remission, a direction we now explicitly propose for future research.

We performed additional flow cytometry analysis on blood and spleen samples from the in vivo study to track the persistence of human SNK (CD3-CD56+) cells over time. The data are now presented in Figure 4E-G, and described on Page 8 and Lines 276-286.

Transcriptomic Analysis

The RNA-seq findings are summarized descriptively but lack statistical depth (no enrichment P-values).

The absence of a clear biological focus weakens the interpretation.

 No validation of RNA-seq data by qPCR or protein assays undermines reliability.

Answer: Thank you for your comment. Enrichment P-values are now presented in Figure 5I-L, and described on Page 10 and Lines 314-319. Violin plots showing the transcripts per million (TPM) of cytokine and chemokine signaling, immune activation, cytotoxic effector function, and signaling by interleukin-related genes in PBNK, and SNK cell groups (n=3 donors).

We thank the reviewer for this insightful comment. We acknowledge that direct experimental validation, such as qPCR or Western blot, would provide additional support for our findings. However, the primary objective of our study was to conduct an unbiased transcriptomic profiling to identify a comprehensive set of genes and pathways that are potentially involved in PBNK vs SNK. In this context, we believe that our RNA-seq data, which was generated with high sequencing depth and robust biological replicates, provides a highly reliable and quantitative dataset for identifying differentially expressed genes (DEGs).

To further address the reviewer's concern and strengthen our conclusions without additional experiments, and we have toned down the language regarding causal mechanistic conclusions throughout the manuscript, particularly in the Discussion section. We now more explicitly state that our study "identifies candidate genes and suggests potential mechanisms" that warrant future investigation, rather than claiming definitive proof. (Page 13 and Line 380).

Tables and Figures

Figures lack error bars and sample sizes, preventing assessment of data variability.

Figure 2 legend and labeling do not match the text; some subpanels are missing or ambiguous in the supplied figures.

Figures 4C–D show marginal survival benefit (19 vs. 17 days), which is not biologically meaningful despite statistical significance.

P-values not p-value.

 Answer: Thank you for your comment. Error bars and sample sizes of figures were added.

The text of Figure 2 was revised in Page 6 and Line 226.

We thank the reviewer for this comment regarding the survival analysis. The primary and more robust evidence of SNK cell efficacy in vivo comes from the significant suppression of tumor bioluminescence observed up to Day 10 (Figure 4B). This early and potent anti-tumor effect provides a clear and direct demonstration of biological activity. Furthermore, to better understand the kinetics underlying this response, we performed longitudinal tracking of SNK cells. As now detailed in Figure 4E-G, we found that the SNK cell population peaked in the peripheral blood around Day 7 and subsequently declined rapidly. This kinetic profile strongly suggests that the potent tumor suppression up to Day 10 was driven by this wave of effector cells, and the subsequent tumor relapse correlates with their contraction. Therefore, while the sample size (n=5) was sufficient to demonstrate this clear pattern of initial efficacy and to justify future studies, we acknowledge that achieving a more profound survival benefit will likely require a multi-dosing strategy to maintain effective SNK cell levels in vivo. We have now explicitly stated this important limitation and future direction in the Discussion section (Page 12 and Lines 376-377).

p-value was revised with P-values in text.

Discussion

The discussion repeats result rather than critically interpreting them.

It lacks mechanistic insight into how CD16/CD137 co-stimulation mechanistically differs from cytokine-only activation.

   Answer: Thank you for your comment. The discussion were added in Page 12 and Lines 352-358, and emphasized the ability of IL-2 and IL-15 to proliferate and antibodies CD16 and CD137 to activate.

The late-phase tumor regrowth is attributed to reduced persistence but no data support this claim (e.g., lack of persistence or exhaustion marker analysis).

Answer: Thank you for your comment. The results of the fate SNK in NSG mice were added in Figure 4E-G, and discussion was added in described on Page 12 and Lines 375-277.

The comparison to existing NK cell expansion systems (e.g., feeder-based or CAR-NK) is missing, leaving unclear what the true innovation is.

Answer: Thank you for your suggestion. we have revised the introduction to better articulate the specific advantages of our CD16/CD137 dual-antibody approach. As the reviewer rightly points out, we did not perform a direct, side-by-side comparison with feeder cell-based or cytokine cocktail methods in this study. Therefore, we refrain from claiming superiority in terms of NK cell expansion efficiency. Instead, the primary advantages of our protocol lie in its simplicity, cost-effectiveness, and enhanced safety profile. The content of introduction were revised and added on Page 2 and Lines 74-93.

Potential limitations (single donor, lack of heterogeneity, single tumor model) are not acknowledged.

To optimistic translational claims are made without preclinical safety or persistence data.

Answer: Thank you for your suggestion. we have deleted the translational claims.

Future directions are missing. Briefly suggest next steps, such as investigating NK cell persistence, underlying activation mechanisms, and validation in additional in vivo models to strengthen the study’s translational impact.

Answer: Thank you for your suggestion. Future directions were added in Page 13 and Lines 393-395.

Conclusions

The conclusion is currently embedded within the Discussion. If journal formatting requires a distinct “Conclusions” section, this paragraph (Lines 316-323) should be separated under that heading for clarity and consistency.

 Answer: Thank you for your comment. The conclusion has been revised in Page 13 and Line 397.

References

References needs to be 10 years back not more (from 2015 to 2025).

Several citations are duplicated (e.g., #10 and #11 are identical).

Many key references are dated (2001–2019), omitting recent advances (2023–2025) in NK cell manufacturing and persistence strategies.

No citation of clinical trial data or GMP-compliant expansion literature.

Please check journal guidelines for reference writing. References are formatted inconsistently (minor typographical issues in author initials and journal titles).

Answer: Thank you for your comment. The references has been revised in Pages 14-15 and Lines 432-461, 481-513.

Reviewer 4 Report

Comments and Suggestions for Authors

The manuscript presents a compelling strategy for the in vitro activation and large-scale expansion of Natural Killer cells derived from peripheral blood mononuclear cells using combined anti-CD16 and anti-CD137 antibody stimulation, as well as IL-2 and IL-15 under serum-free conditions. The resulting Super NK cells demonstrated robust proliferation and significantly enhanced anti-tumor functions both in vitro and in vivo. The study is scientifically sound, highly relevant, and the data is generally well-supported. There are few weaknesses that should be handled carefully, and addressed appropriately by the authors. I summarized them at the end of the text.

Major Findings and Strengths:

  1. Robust Expansion Method: The combination of anti-CD16 and anti-CD137 stimulation resulted in a highly pronounced proliferative response, achieving over 819.2-fold expansion of NK cells within 15 days of culture, which significantly exceeded single antibody stimulation methods. The resulting SNK cells maintained high purity (>98%).
  2. Enhanced Functionality: SNK cells acquired a mature, functionally active phenotype, characterized by the significant upregulation of key activating receptors such as CD69, NKG2C, and NKG2D.
  3. Superior Cytotoxicity: SNK cells exhibited significantly increased cytotoxicity against all tested tumor cell lines (HK-1, DU145, and K562) compared with Peripheral Blood NK (PBNK) cells. Notably, SNK cells achieved approximately 99.14% lysis of K562 cells at an E:T ratio of 2:1 in vitro. This enhanced killing was associated with significantly higher secretion of effector molecules IFN-γ and Granzyme B (Gzm B).
  4. In Vivo Efficacy: Adoptive transfer of SNK cells resulted in significant, dose-dependent tumor suppression and prolonged survival in the K562-engrafted NSG mouse model.
  5. Transcriptomic Insights: RNA sequencing confirmed that the activation method induced broad transcriptional reprogramming, upregulating genes associated with proliferation, immune activation (e.g., CD48, CD70), cytokine signaling (e.g., CCR2, CXCR6), and cytotoxic effector function (e.g., GZMB, PRF1, NCR2, NCR3).

Required minor revisions:

  1. Discussion of Inhibitory Receptors: The results section notes that inhibitory receptors, including CD94-NKG2A, Tim-3, and CD96, were significantly upregulated in SNK cells. While this acquisition is characteristic of a mature phenotype, it may also imply potential susceptibility to inhibitory signals in vivo. Although the Discussion mentions the activated phenotype, clarification on the potential functional implication of the upregulated inhibitory receptors in the context of persistence and anti-tumor function would strengthen the discussion, especially considering the observed tumor regrowth after Day 13.
  2. Persistence Limitation: The authors accurately highlight the limitation of rapid tumor proliferation observed after day 13 and suggest that this is likely due to reduced persistence or viability of the transferred SNK cells. While the conclusion appropriately calls for additional strategies to enhance persistence, expanding the discussion briefly on potential mechanisms for improving SNK cell persistence (e.g., genetic engineering, cytokine support) would enhance the translational scope of the manuscript.

Author Response

The manuscript presents a compelling strategy for the in vitro activation and large-scale expansion of Natural Killer cells derived from peripheral blood mononuclear cells using combined anti-CD16 and anti-CD137 antibody stimulation, as well as IL-2 and IL-15 under serum-free conditions. The resulting Super NK cells demonstrated robust proliferation and significantly enhanced anti-tumor functions both in vitro and in vivo. The study is scientifically sound, highly relevant, and the data is generally well-supported. There are few weaknesses that should be handled carefully, and addressed appropriately by the authors. I summarized them at the end of the text.

Answer: The authors especially thank reviewer for your sincere and responsible suggestions, which have helped our article improve a lot.

Major Findings and Strengths:

  1. Robust Expansion Method: The combination of anti-CD16 and anti-CD137 stimulation resulted in a highly pronounced proliferative response, achieving over 819.2-fold expansion of NK cells within 15 days of culture, which significantly exceeded single antibody stimulation methods. The resulting SNK cells maintained high purity (>98%).

Answer: Thank you for your suggestion. The content of was revised on Page 5 and Lines 193-197.

  1. Enhanced Functionality: SNK cells acquired a mature, functionally active phenotype, characterized by the significant upregulation of key activating receptors such as CD69, NKG2C, and NKG2D.

Answer: Thank you for your suggestion. The content of was revised on Page 6 and Lines 215-216.

  1. Superior Cytotoxicity: SNK cells exhibited significantly increased cytotoxicity against all tested tumor cell lines (HK-1, DU145, and K562) compared with Peripheral Blood NK (PBNK) cells. Notably, SNK cells achieved approximately 99.14% lysis of K562 cells at an E:T ratio of 2:1 in vitro. This enhanced killing was associated with significantly higher secretion of effector molecules IFN-γ and Granzyme B (Gzm B).

Answer: Thank you for your suggestion. The content of was revised on Pages 6-7 and Lines 237-242, 258-261.

  1. In Vivo Efficacy: Adoptive transfer of SNK cells resulted in significant, dose-dependent tumor suppression and prolonged survival in the K562-engrafted NSG mouse model.

Answer: Thank you for your suggestion. The content of was revised on Page 8 and Lines 276-286.

  1. Transcriptomic Insights: RNA sequencing confirmed that the activation method induced broad transcriptional reprogramming, upregulating genes associated with proliferation, immune activation (e.g., CD48, CD70), cytokine signaling (e.g., CCR2, CXCR6), and cytotoxic effector function (e.g., GZMB, PRF1, NCR2, NCR3).

Required minor revisions:

Answer: Thank you for your suggestion. The content of was revised on Page 10 and Lines 314-319.

Required minor revisions:

  1. Discussion of Inhibitory Receptors: The results section notes that inhibitory receptors, including CD94-NKG2A, Tim-3, and CD96, were significantly upregulated in SNK cells. While this acquisition is characteristic of a mature phenotype, it may also imply potential susceptibility to inhibitory signals in vivo. Although the Discussion mentions the activated phenotype, clarification on the potential functional implication of the upregulated inhibitory receptors in the context of persistence and anti-tumor function would strengthen the discussion, especially considering the observed tumor regrowth after Day 13.

Answer: Thank you for your suggestion. We agree that the functional consequence of co-upregulation of activating and inhibitory receptors is complex and can be ambiguous. While we did not perform direct functional assays for specific receptor-ligand interactions (e.g., NKG2A blockade), we have analyzed our data to distinguish between an "inhibitory" and a "highly activated" phenotype based on several functional correlates. Furthermore, The primary and more robust evidence of SNK cell efficacy in vivo comes from the significant suppression of tumor bioluminescence observed up to Day 10 (Figure 4B). This early and potent anti-tumor effect provides a clear and direct demonstration of biological activity. Furthermore, to better understand the kinetics underlying this response, we performed longitudinal tracking of SNK cells. As now detailed in Figure 4E-G, we found that the SNK cell population peaked in the peripheral blood around Day 7 and subsequently declined rapidly (Page 8 and Lines 276-286).

  1. Persistence Limitation: The authors accurately highlight the limitation of rapid tumor proliferation observed after day 13 and suggest that this is likely due to reduced persistence or viability of the transferred SNK cells. While the conclusion appropriately calls for additional strategies to enhance persistence, expanding the discussion briefly on potential mechanisms for improving SNK cell persistence (e.g., genetic engineering, cytokine support) would enhance the translational scope of the manuscript.

Answer: Thank you for your suggestion. We performed additional flow cytometry analysis on blood and spleen samples from the in vivo study to track the persistence of human SNK (CD3-CD56+) cells over time. The data are now presented in Figure 4E-G, and described on Page 8 and Lines 276-286.

Round 2

Reviewer 1 Report

Comments and Suggestions for Authors

The authors took all my comments into account. The article can be accepted in its current form.

Reviewer 3 Report

Comments and Suggestions for Authors

none